# Pregnancy Management in HIV Viral Controllers: Twenty Years of Experience

**DOI:** 10.3390/pathogens13040308

**Published:** 2024-04-10

**Authors:** Charlotte-Eve S. Short, Laura Byrne, Aishah Hagan-Bezgin, Rachael A. Quinlan, Jane Anderson, Gary Brook, Okavas De Alwis, Annemiek de Ruiter, Pippa Farrugia, Sarah Fidler, Eleanor Hamlyn, Anna Hartley, Siobhan Murphy, Heather Noble, Soonita Oomeer, Sherie Roedling, Melanie Rosenvinge, Luciana Rubinstein, Rimi Shah, Selena Singh, Elizabeth Thorne, Martina Toby, Brenton Wait, Liat Sarner, Graham P. Taylor

**Affiliations:** 1Department of Infectious Disease, Imperial College London, London W2 1PG, UK; 2Imperial College NIHR BRC, Imperial College London, London W2 1NY, UK; 3Imperial College Healthcare NHS Trust, London W2 1NY, UK; 4School of Medicine, St Georges, University of London, London SW17 0RE, UK; 5St. George’s University Hospitals NHS Trust, London SW17 0RE, UK; 6School of Medicine, University of Liverpool, Liverpool L69 3GE, UK; 7Homerton Healthcare NHS Foundation Trust, London E9 6SR, UK; 8London North West University Healthcare NHS Trust, Harrow HA1 3UJ, UK; 9Guy’s and St Thomas’ NHS Foundation Trust, London SE1 7EH, UK; 10ViiV Healthcare, Brentford TW8 9GS, UK; 11Royal Free London NHS Foundation Trust, London NW3 2QG, UK; 12Barts Health NHS Trust, London E1 1BB, UK; 13Leeds University Teaching Hospital NHS Trust, Leeds LS1 3EX, UK; 14Central and North West London NHS Foundation Trust, London NW1 3AX, UK; 15Lewisham and Greenwich NHS Trust, London SE13 6LH, UK

**Keywords:** elite HIV viral controller, pregnancy, antiretroviral therapy, obstetric management, HIV-1 vertical transmission

## Abstract

(1) Background: The evidence base for the management of spontaneous viral controllers in pregnancy is lacking. We describe the management outcomes of pregnancies in a series of UK women with spontaneous HIV viral control (<100 copies/mL 2 occasions before or after pregnancy off ART). (2) Methods: A multi-centre, retrospective case series (1999–2021) comparing pre- and post-2012 when guidelines departed from zidovudine-monotherapy (ZDVm) as a first-line option. Demographic, virologic, obstetric and neonatal information were anonymised, collated and analysed in SPSS. (3) Results: A total of 49 live births were recorded in 29 women, 35 pre-2012 and 14 post. HIV infection was more commonly diagnosed in first reported pregnancy pre-2012 (15/35) compared to post (2/14), *p* = 0.10. Pre-2012 pregnancies were predominantly managed with ZDVm (28/35) with pre-labour caesarean section (PLCS) (24/35). Post-2012 4/14 received ZDVm and 10/14 triple ART, *p* = 0.002. Post-2012 mode of delivery was varied (5 vaginal, 6 PLCS and 3 emergency CS). No intrapartum ZDV infusions were given post-2012 compared to 11/35 deliveries pre-2012. During pregnancy, HIV was detected (> 50 copies/mL) in 14/49 pregnancies (29%) (median 92, range 51–6084). Neonatal ZDV post-exposure prophylaxis was recorded for 45/49 infants. No transmissions were reported. (4) Conclusion: UK practice has been influenced by the change in guidelines, but this has had little impact on CS rates.

## 1. Introduction

HIV-1 viral controllers (also known as elite controllers) are defined by their ability to spontaneously control HIV viraemia to below the level of quantification (LLoQ) of commercial viral load assays for prolonged periods in the absence of antiretroviral therapy (ART) [1]. Viral controllers are reported to comprise less than 1–5% of individuals, varying between cohorts with an association with both female gender and Black heritage backgrounds [2,3,4,5,6]. A recent observational arm within a randomised control trial (RCT) of Option B+ in pregnant Uganda women revealed a high prevalence (21%) of plasma viral concentrations < 400 copies/mL at baseline with no reported prior ART, with 6% having a viral load < 20 copies/mL [7].

The mechanisms of viraemic control are incompletely understood but include host genetics either rendering host T cells difficult to infect via coreceptor mutation or through enhanced antigen presentation on infected T cells improving targeted host immune surveillance [2,8,9,10]; enhanced immunological control through effective polyfunctional CD8 responses [1,11] and broadly neutralising antibody responses [12]; as well as reduced viral reservoirs and enhanced deep latency through integration into transcriptional deserts [13]. Due to this immunological heterogeneity, there is no universally accepted definition of viral controllers nor consensus on optimum management, including during pregnancy. Over time, some of these individuals experience loss of virological control, immune activation, and possibly adverse outcomes [14,15,16,17]. How pregnancy impacts this is unknown, but the relative immune suppression associated with pregnancy makes this a unique situation.

Maternal viral load during pregnancy and delivery is the single most important predictor of vertical transmission [18]. Unlike sexual transmission, where a sustained undetectable plasma viral load renders the individual non-infectious [19,20,21], in pregnancy, transmission may still rarely occur even when the plasma viral load at delivery is below the limit of detection, even in the absence of other clear complicating factors [22,23,24,25,26]. In the seminal PACT076 study of zidovudine monotherapy (ZDVm) in pregnancy, one transmission occurred despite a maternal viral load below the LLoQ (<400 copies/mL) prior to commencing ART [27]. The benefit of ZDVm in PACT076 was seen at all levels of maternal viral load, resulting in the conclusion that in addition to reducing viral load, ART may also confer another protective mechanism. The benefits of triple ART in dramatically reducing the risk of vertical transmission in late presenters, despite detectable virus late in pregnancy, have since been demonstrated in the DolPHIN-2 randomised control trial [28], suggesting that transplacental pre-loading of the foetus with maternal ART also plays a role.

The specific risk of vertical transmission in pregnant HIV-1 viral controllers is unknown but is likely to be low. Most people with plasma viral suppression on ART are also suppressed in other compartments; however, there is a theoretical risk of discordance, with cases reported of detectable virus in cervicovaginal secretions [29,30]. It is possible that any persistent detectable virus, whilst on ART, in the vaginal compartment is also not the result of viral replication, rather shedding of virions from cellular clones [31]. This impact of any viral shedding on risk transmission in viral controllers is probably minimal; however, it has been shown that free and cell-associated virus in cervicovaginal fluid are risk factors for vertical transmission [32]. The potential for changing viral dynamics in the face of the relative immunosuppression of pregnancy is also a theoretical consideration.

There are no clinical trials or observational studies investigating the optimum management of viral controllers in pregnancy, and the literature is limited to case reports [33]. Prior to a revision of UK British HIV Association (BHIVA) guidelines in 2012, which made specific recommendations for viral controllers, previous guidelines recommended consideration of ZDVm for individuals with viral loads of <10,000 copies/mL together with PLCS with intravenous ZDV infusion at delivery [34]. In 2012, BHIVA guidelines recommended that viral controllers be managed with either ZDVm or triple ART, aiming for vaginal delivery [35]. From 2018, the guidelines referred to BHIVA adult treatment guidelines, which recommend lifelong triple ART for all [36,37].

The aim of this work is to increase the available evidence by describing the management experience and neonatal outcomes of a series of pregnancies women with HIV viral control managed over the past three decades in the UK.

## 2. Materials and Methods

Viral controller pregnancy cases were identified through treating physicians participating in a regional network, where management of HIV in pregnancy is discussed. Viral controllers were defined using the following inclusion criteria: (i) women who were HIV-1 antibody positive (HIV-2 negative), and (ii) a HIV viral load less than 100 copies/mL on at least two consecutive assays either prior to initiating antiretroviral therapy in pregnancy or off ART post-partum in their first captured pregnancy, and (iii) one or more pregnancies leading to live birth. There was no limit to the retrospective data collection period, with the earliest data captured from 1999 onwards.

Clinicians completed a standardised anonymised case record form for each pregnancy using data from electronic and paper clinical records which were then collated and analysed centrally by the London HIV Perinatal Research Group. As an evaluation of management practice, this work was conducted as a clinical governance activity and thus neither individual consent nor formal ethical approval were required. Additional data on infant infection status was sought from the Integrated Screening Outcomes Surveillance Service (previously National Study of HIV in Pregnancy and Childhood) where required.

Data were captured in two time periods—first in 2011 and again in 2021. Analyses were divided by deliveries pre- and post-2012, when concurrent UK guidelines made specific recommendations for management of pregnant viral controllers (including treatment with either ZDVm or triple ART). Demographic, immunological and virological characteristics and perinatal outcomes for women across both time periods were described and compared using the median, Fisher’s exact, and Chi-squared tests in SPSS (version 28 for Macintosh). Missing data were excluded from analyses and total numbers available for description of each parameter are represented in results.

## 3. Results

### 3.1. Demographics

A total of 49 singleton pregnancies in 29 women that resulted in live birth were identified across 13 HIV treatment centres; 13 of the women had more than 1 pregnancy recorded, 4 of whom had deliveries in both time periods. Year of delivery ranged over 1999–2021, 35 before 2012 and 14 during or post-2012. Median maternal age at antenatal booking was 31 years, range 18–40 years. Maternal age in pregnancies post-2012 was higher than pre, 33 years vs. 31 years (*p* = 0.05); see Table 1. Year of first HIV diagnosis ranged from 1996 to 2016. HIV was diagnosed during the reported pregnancy in 35% of cases (17/49), with a trend towards a higher proportion diagnosed during pregnancy pre-2012 compared to post (43% (15/35) vs. 14% (2/14), *p* = 0.10). 73% of women (21/29) were born in Sub-Saharan Africa and 14% (4/29) in the UK. Transmission route was heterosexual sex for all but one case who also had a history of injecting drug use. Comorbidity was reported in 23/49 (47%) pregnancies. Women delivering post-2012 were more likely to have comorbidity than pregnancies pre-2012 (86% (12/14) vs. 32% (11/35), *p* = 0.001); see Table 1. Co-morbidities and complications included asthma, hyperemesis, pre-eclampsia, hypertension, type 2 diabetes, iron deficiency anaemia, immune-mediated thrombocytopenia, latent *Mycobacterium tuberculosis* (receiving treatment with Rifampicin and Isoniazid), hepatitis B and hepatitis C, herpes simplex virus and depression. Fourteen pregnancies were in primiparous women, more pre-2012: 34% (12/35)) vs. post: 14% (2/14), *p* = 0.29. None of the women had been treated with ART prior to the first reported pregnancy.

### 3.2. Baseline Immune and Virologic Parameters

Median CD4 count at diagnosis was 604 cells/µL, range 350–997. CD8 values at diagnosis were available for 12 cases and the median was 542 cells/µL range 250–1128. CD4/CD8 ratio at diagnosis were available for 13 cases with a median of 1.09 range 0.66–2.10. Median CD4 count during the recorded pregnancy was 735 cells/µL, range 306–1227 (missing for 5 cases). CD4/CD8 ratio during pregnancy was available for 21 cases with a median of 1.3 (range 0.4–2.1). CD4/CD8 ratios < 1 in pregnancy were recorded for 6/21 cases (29%).

All women had plasma HIV RNA measurement repeated on an alternative assay and/or had a proviral DNA measured. Measurements of HIV RNA were performed on eight different commercial platforms with LLoQ that ranged from 20 to 400 copies/mL; see Appendix A. Several of these assays can detect plasma virus below the LLoQ and were reported as such. Four women had no detectable plasma HIV RNA, two with repeat pregnancies; see Appendix A. Maternal proviral DNA results were available in 20/29 women, and DNA targets were amplified in 12 of these women. Of the four women with no detectable RNA copies detectable throughout pregnancy, one had proviral DNA amplified in three pregnancies, one initially had detectable HIV proviral DNA in her first reported pregnancy and then the proviral DNA was not amplified in the second and in the other two pregnancies no proviral DNA targets where amplified.

### 3.3. Ante and Perinatal Maternal Management

ART was used in all pregnancies. Figure 1 shows ART and mode of delivery of pregnancies by year of delivery. Pre-2012 pregnancies were predominantly (80%) managed with ZDVm (28/35) compared to 29% (4/14) post-2012, *p* = 0.002.

Post-2012 triple ART was initiated or continued in more pregnancies—72% (10/14) compared to 20% (7/35) of cases pre-2012, *p* = 0.002. Of the seventeen pregnancies in which triple ART was used, seven cases received protease inhibitor (PI) based regimens (four pre-2012), two non-nucleoside reverse transcriptase inhibitors (NRTI) based regimens, two a CCR5 receptor antagonist-based regimens, four cases took triple NRTI regimens (three pre-2012, including the woman being treated for latent TB) and two an integrase strand transfer inhibitor-based regimen; see Figure 2A.

One woman had two pregnancies pre-2012, where she received ZDVm, and two post-2012, where she conceived on triple ART. The remaining initiated ART during pregnancy at a median gestational age of 28 weeks for ZDVm (range 20–34) and triple ART at 25 weeks (range 7–33), *p* = 0.86.

During pregnancy, plasma HIV RNA was detected (>50 copies/mL) in 14/49 pregnancies (29%) (range 51–6084). The trimester of detectable HIV RNA varied with no consistent trend in relation to ART treatment; see Appendix A. Thirty-five (72%) women had a plasma viral load < 50 copies throughout pregnancy, one of which had a measurement between 20 and 49 copies/mL (29 copies/mL), and three of whom for which all were results were reported to be <20 copies/mL.

Forty-seven (96%) cases had an undetectable HIV plasma RNA concentration at delivery (<50 copies/mL). One woman, who was diagnosed in pregnancy, received triple ART containing a PI from the third trimester, and who delivered vaginally, had a delivery plasma RNA measurement of 205 copies/mL (only available post-delivery). All her preceding results were <50 copies/mL. The other women who took ZDVm had detectable plasma virus through out pregnancy and a viral load of 54 copies/mL on the day of pre-labour caesarean section (PLCS); her post-partum viral load off ART was 247 copies/mL, whereas all measurements were <50 copies/mL in her previous pregnancy; see Appendix A.

Intravenous AZT was administered in 11/35 deliveries pre-2012, but not used post-2012. Mode of delivery was PLCS in 25/35 (72%) cases pre-2012 and 7/35 (20%) were vaginal deliveries; see Figure 1. Post-2012 mode of delivery was varied (5 vaginal, 6 PLCS and 3 Emergency CS (EmCS)); see Figure 1 and Figure 2B. Indication for PLCS was prevention of vertical transmission in combination with ZDVm in 24/30 and previous C-section with triple ART for 6/30 planned C-sections. Proportionally, there were more EmCS post-2012 (22% versus 12% *p* = 0.39). Median gestation at delivery was 38 weeks (range 32–41). Three infants were born < 37 weeks gestation, all pre-2012, giving a preterm birth rate of 7%; two of these were born by EmCS for PPROM at 32 weeks and foetal distress at 34 weeks, respectively.

### 3.4. Post-Partum and Neonatal Management

Neonatal ZDV post-exposure prophylaxis given for 28 days was recorded for 44/49 (90%) infants. Of the remaining infants, one did not receive any post-exposure prophylaxis, one received 14 days of ZDV, and for three cases the records were not available. Maternal ART was stopped post-delivery in all pregnancies pre-2012 and in 10/14 cases post-2012, *p* = 0.005. In the cases where ART was continued post-partum, all received triple ART on a long-term basis.

There were no reports of breastfeeding pre-2012. Post-2012, two women chose to breast feed—both took triple ART in pregnancy and continued post-partum. Infant HIV status (12 week DNA/RNA status and or antibody at 18–24 months) was available for all except one case. Infant antibody status was recorded and negative for 40/49 births and negative 12 week HIV DNA or RNA PCR was available for 41/49 cases. For a summary of cases in this series, see Appendix A. No transmissions have been reported in 48 cases—for one case, no infant testing results were available.

## 4. Discussion

To the best of our knowledge, this is one of the first case series examining the management and outcomes of pregnancy in people with HIV (PWH) who fulfil a definition of HIV viral controllers, with a plasma HIV VL < 100 off ART pre- or post-pregnancy. There has been considerable variation in the management of these individuals across the twelve HIV treatment centres, which to a certain extent illustrates both the lack of evidence base and the evolution in UK guidelines for the management of pregnancy in PWH during the study period [34,35,36,38]. In this case series, 80% of women were treated with ZDVm pre-2012, and 25% post-2012, which remains higher than observed in national UK surveillance figures. In 1999, 20% of pregnant women in the UK and Ireland received ZDVm overall, declining to 4.4% in 2006, 2% in 2011 and only 0.6% of all pregnancies in 2018 had any treatment with ZDVm [39,40]. The reduction in the use of intrapartum intravenous ZDV over the study period is also likely due to a better understanding that this intervention confers limited additional benefit in reducing vertical transmission within the context of a maternal plasma viral load < 1000 copies/mL [41]. The reduction in the use of oral (and intravenous) ZDV across the study period is in keeping with sequential changes in national BHIVA pregnancy management guidelines.

As expected from published data on the effectiveness of ZDVm with PLCS in women with detectable HIV RNA pre-treatment [27,42,43], there were no transmissions with the use of this approach in this case series. It is therefore likely that the increasing choice of triple ART over ZDVm pre-2012 was to ensure PWH could aim for vaginal delivery, since all women in this period treated with ZDVm were advised to deliver by caesarean section [34]. Post-2012 the increasing numbers of vaginal deliveries and VBAC alongside increasing triple ART use is likely to have been driven by concurrent UK BHIVA guidance published in 2012 recommending viral controllers could aim for vaginal delivery irrespective of type of ART and specifying triple ART (including triple NRTI) as a treatment option [35]. In 2015, WHO and UK adult treatment guidance recommended that all people living with HIV should receive ART, regardless of CD4 count or clinical status [37,44]. Aligned with these updates, 2018 UK BHIVA pregnancy guidance recommended that viral controllers be managed according to the same principle, including initiating and continuing ART lifelong, not only for benefits to pregnancy, but to clinical outcomes and reduction in HIV transmission [36]. The continued use of ZDVm and triple NRTI post-2018 likely reflects patients’ previous experience and confidence with these interventions, and counselling on limited evidence base for triple ART in the context of a plasma viral load <50 copies/mL. Concern about higher rates of preterm birth (PTB) in people treated with triple-combination therapies, especially protease inhibitor-based combinations, may also have influenced clinician and maternal choices.

Ongoing high rates of PLCS may be the result of weighing the balance of the risk of uterine rupture with vaginal birth after repeated C-section (VBAC). Emergency C-section rates increased across the time periods from 11% to 27% but this observation did not reach statistical significance, probably due to the small sample size. The higher proportion of unscheduled C-sections is likely to be a consequence of wider patient choice to include vaginal delivery and therefore some women who planned to have a VBAC, were ultimately converted to emergency C-Section during delivery. This figure is similar to previous UK data collected between 2013 and 2014 demonstrating an EmCS rate of 28% in women who were eligible for and planned vaginal delivery based on their plasma HIV copy number [45]. It is also similar to the increasing rates of EmCS seen in the UK and Ireland from 1999 (17%), rising to 22.9% in 2006 [39], from whence it has remained fairly stable to the current time [46,47,48].

The PTB rate of 7% is comparable to the 2021 national rate for England and Wales (8%) [49] and half the rate observed in PWH nationally which is stable at 12% since 2015 [47,48]. The rate of PTB in viral controllers is similar to lower PTB rate observed in pregnant PWH treated with ZDV monotherapy compared to those treated with triple ART over comparable time periods, both at a UK leading tertiary centre (1996–2010) [50] and nationally (1990 and 2006) [51]. This observation is particularly interesting given PTB is an inflammatory syndrome [52], particularly in the context of HIV [53,54], and the lower PTB rates could hypothetically reflect dampened immune activation in viral controllers which may in turn reduce the risk of this complication.

Despite a lack of evidence for management of HIV in people with spontaneous viral control, current 2022 UK adult treatment guidance recommends that pregnancy is a scenario where ART should be used due to uncertainty over the impact of the relative immune suppression of pregnancy on viral control and therefore the potential risk of transmission [55]. Indeed, in this series, we identified detectable viraemia in 29% of pregnancies, indicating that viral rebound can occur in some viral controllers during pregnancy. These current adult treatment guidelines highlight that for viral controllers, who maintain plasma viral concentrations <50 copies/mL without ART and have normal CD4 counts, in the absence of other co-morbid conditions for which ART use is indicated, the benefits of combination ART remain uncertain and recommend that such cases are referred to the UK national NHS clinical service (imperial.idris@nhs.net). All patients referred to this clinical service undergo extensive investigations to further characterise their viral control [56]. A more rigorous definition is applied to include molecular positive or negative features as assessed by proviral DNA and ultrasensitive viral load assays, whilst the presence of immune activation is also assessed. Treatment recommendations in non-pregnant viral controllers are limited to those with evidence of molecular positivity, evidence of immune activation or other non-infectious morbidity in which HIV infection may be an unfavourable factor, e.g., cardiovascular disease and cancer.

All cases in this pregnancy series of viral controllers had viral load measurement repeated on another platform with a variety of lower limits of quantification and detection and heterogeneity in how this was reported, e.g., less than the LLoQ, zero, or not detected; see a summary of the assays used in Appendix A. Maternal proviral DNA results were available for two-thirds of cases and in two women undergoing RNA quantification on a platform that reported lower limit RNA concentrations of zero, DNA was also not detected. One woman had an evolving DNA result, with primers amplifying HIV in her first reported pregnancy and not in the second. Sensitivity and specificity of DNA Primers will have improved over the study period, so it is difficult to make any inference from these results on viral control or reservoir. Of the 21 pregnancies for which there was a CD4/CD8 ratio available, the majority were within normal range >1; however 6 cases had ratios <1, which would be an indication alone for ART in spite of pregnancy. In addition, several women also had comorbidities that would be an indication for ART, e.g., hepatitis B coinfection.

The majority of cases stopped ART post-partum, even in the later time period, which is in contrast to current trends of increasing women conceiving on ART and national guidance to continue ART post-partum [36,40,45]. This observation is likely to reflect patient choice, in conjunction with counselling from their treating physician on the risks and benefit of continued ART in people who maintain viral control off ART. It may be safe and feasible for such women to interrupt ART post-delivery if they wish to but ideally this should be managed in a carefully controlled specialist centre. Reasons to continue triple ART in this series included both breast feeding and the presence of treatment indicators comorbid conditions, e.g., hepatitis B and hypertension. Interestingly, pregnancies were more likely to be in women with comorbid conditions in the later time period, which is likely to reflect both that these women were slightly older and that a higher proportion were repeat pregnancies in individuals with significant medical history, some of which were indicators for ART.

Treatment decisions in pregnant HIV viral controllers should consider transmission risks including infant feeding decisions, the potential adverse effects of ART, comorbidities and patient choice. The high rate of viral rebound and prevalence of reduced CD4/CD8 ratio are supportive of the recommendation that pregnant women with viral control should receive ART. Individualised care is needed for all viral controllers who are pregnant or planning pregnancy, both for clinical assessment and ART counselling. This could include consideration of referral to an expert centre such as the UK national NHS clinical service (imperial.idris@nhs.net). Although not in current UK BHIVA pregnancy management guidance, ZDVm is occasionally used at a PWH’s request, often in the context of the previous use of this management option in pregnancy, with the knowledge that it has been established that it reduces peripartum vertical transmission rates in women with low or undetectable viral loads [22,23]. Triple ART is not without potential adverse effects, initiating PI- based triple ART in pregnancy has historically been associated with PTB, as well as other specific drug-related toxicities [57,58]. Recent metanalyses and systemic review of Sub-Saharan African cohorts and RCT data have shown that preconception and antenatal triple ART are associated with PTB, low birth weight and small for gestational age babies in women living with HIV compared to those receiving ZDVm and to uninfected women [59]. It is possible that the continued association of triple ART with adverse birth outcomes may be drug specific, with another large meta-analyses from the same group demonstrating that compared with ZDVm, non-nucleoside reverse transcriptase inhibitor-based triple ART was associated with an increased risk of PTB, LBW, and stillbirth, and PI-based triple ART with PTB [60]. Increasing safety data for ART in pregnancy has resulted in more ART choices for pregnant women including Integrase Stand Transfer Inhibitors, with Dolutegravir now a preferred third drug in many international pregnancy guidelines due to its efficacy, reassuring safety profile and fewer drug interactions [28,61,62].

## 5. Conclusions

Viral controllers are rare but are individuals for which a case can be made for avoidance of ART in certain very specific situations. This case series, the largest of its kind, highlights that ART and obstetric management of viral controllers in pregnancy were heterogenous and evolved over time. The high rate of viraemia during pregnancy, although at low copy numbers, is supportive of the potential for detectable viraemia and the use of ART in these cases. Some women and clinicians chose to use more historic less intensive regimes such as ZDVm, presumably to minimise potential risks from triple ART, such as drug toxicity and PTB. Many women planned for vaginal delivery and the majority stopped ART post-partum. Few breast fed; however, this is an absolute indication for triple ART and for continuation through the breastfeeding period. Although total numbers are small, in part due to the rarity of this patient group, it is reassuring that there were no vertical transmissions reported in this series regardless of approach taken. We look forward to prospective data from surveillance studies in the future.

## Figures and Tables

**Figure 1 pathogens-13-00308-f001:**
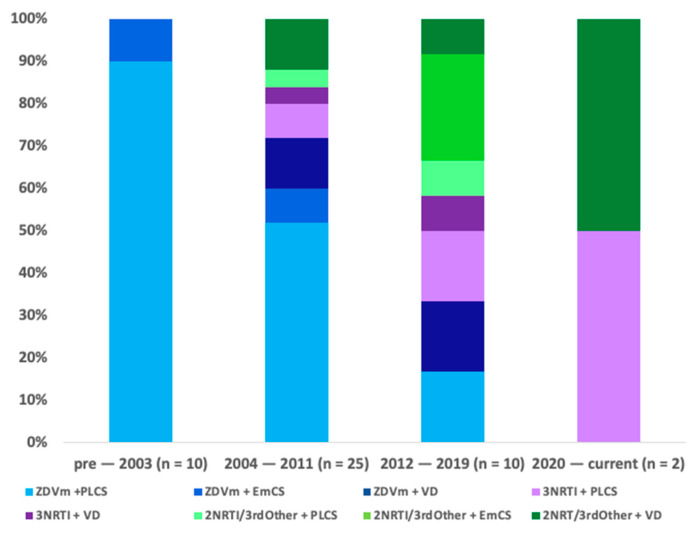
ART and mode of delivery of pregnancies by 8-year period of delivery.

**Figure 2 pathogens-13-00308-f002:**
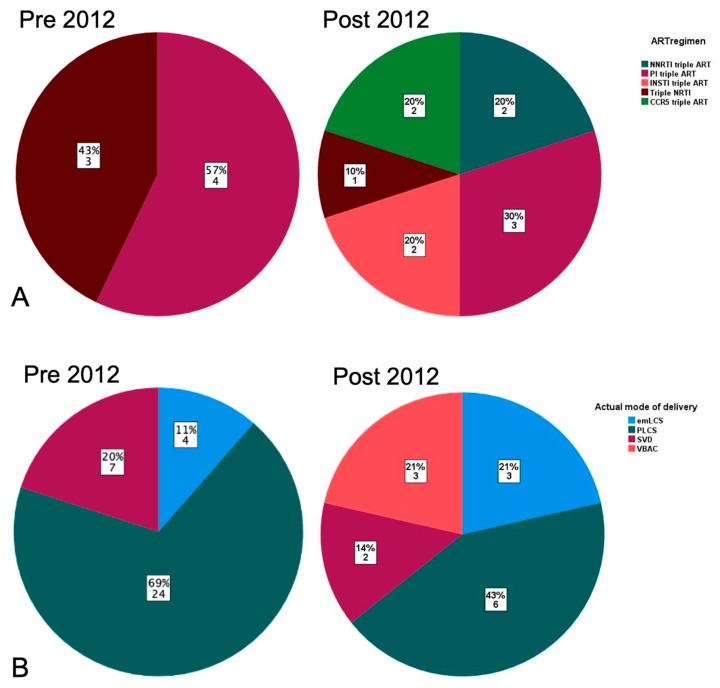
(**A**) Pie chart to show triple ART regimen class by third agent and time period; (**B**) pie chart to show mode of delivery by time period.

**Table 1 pathogens-13-00308-t001:** Summary of baseline demographics of women and pregnancies.

Live Birth	Pre-2012 *n* = 35	Post-2012 *n* = 14	*p* Value
**Median maternal age (years (range))** **Missing (n)**	31 (18–37) 3	33 (24–40) 5	0.05
**Diagnosed with HIV in current pregnancy (*n* (%))**	15 (43)	2 (14)	0.10
**Region of birth** **Europe** **Southeast Asia** **Sub-Saharan Africa** **Caribbean** **Missing (*n*)**	women = 23 5 1 16 0 1	women = 9 2 0 6 1 0	0.53
**Co-morbidity (%)** **Hepatitis B (*n* = 2)** **Hepatitis C (*n* = 1)** **Latent TB** **HTN/GDM**	11 (32) 1 2 1 5	12 (86) 1 0 0 2	0.001
**Para (*n* (%))** **0** **1+**	12 (34) 23 (66)	2 (14) 12 (86)	0.29

## Data Availability

Details of cases for future metanalyses are presented in Appendix A; for further details beyond these data, please contact the corresponding author.

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
