# Peer review of "Pregnancy Management in HIV Viral Controllers: Twenty Years of Experience"

_pathogens, 2024, doi:10.3390/pathogens13040308_

Round 1
Reviewer 1 Report
Comments and Suggestions for Authors
The following article titled "Pregnancy management in HIV viral controllers: twenty years 2 of experience" discusses the impact of change in guidelines in HIV controllers in pregnancy. The study in essence is comparing the difference in treatment of HIV controllers in pregnancy with ZDVm or ART. The study is well written and talks about a very relevant topic. However, I would like the authors to comment on the difference in maternal age in their groups as maternal age is linked with complications during pregnancy. Also was there any difference in women with more than 1 pregnancy? can the authors comment on any impact on the neonates born from these women?
Author Response
Thank you for your supportive review.
We agree that we should add a comment on the increased prevalence of comorbidity in the later part of the study to the discussion with reference to the difference in maternal ages and greater proportion of repeat pregnancies in the later period. We also noted in doing so there was an error in table 1 missed on final editing of the previous submitted manuscript that did not reflect the data that was correctly described in results which we have now corrected i.e. 12 not 10 comorbidities in 14 pregnancies.
Although we agree that considering maternal age and comorbidity is important in birth outcome data, due to overall low incidence of perinatal complications across the whole study (zero PTB post-2012, which is to be expected as a 7% rate in the 14 pregnancies = < 1 expected events) means any exploration of impact of age and repeated pregnancies on birth outcomes would be underpowered and inconclusive, so has not been included.
There was no data collected on the long-term follow up of these infants, which we agree would be of interest, thank you for the suggestion and we can consider this for future work.
Reviewer 2 Report
Comments and Suggestions for Authors
The present study describes the management and evolution of HIV-positive women HIV-RNA viral controlers during their pregnancy and delivery. I find it very interesting because there is limited data on this topic.
However, I believe the article is lengthy and cumbersome, and many data presented are not particularly relevant as it is similar to what it is found in HIV-positive pregnant women with suppressed viremia through treatment. For this reason, I suggest summarizing the article.
For ex, introduction and discussion is too long.
Including so many CD4 and CD4/CD8 ratio data doesn't add much value and could be presented in a table format.
Additionally, concerning viral load, it would be interesting to know how long before pregnancy the last determination was made or if it was during the pregnancy before initiating ART.
In my opinion data don proviral DNA are not necesary, as they do not add much value. We don´t really know the value of HIV-DNA in clinical practice.
In my opinion, the most interesting aspect of the article, and what I think should be emphasized more, is that viral rebound was observed in 29% of pregnancies, so as you discuss, viral rebound can ocure during pregnancy in HIV controlers as a result of changes on immune regulation.
As a minor consideration
The first time an abbreviation appears, it needs to be explained.
For example, in the first paragraph, 'RCT' is not explained. In the fifth paragraph, 'ZDVm' is also not defined.
Sincerely
Author Response
Thank you for your supportive review. We believe that the information in the introduction is pertinent to the topic, introducing the prevalence of spontaneous viral control, the proposed mechanisms in the current field, the importance of plasma HIV RNA to risk of vertical transmission and thus why these pregnancies are unique and the sparsity of evidence base to guide ART management in these cases.
We thank the reviewer for highlighting one of the important findings is that viraemia was detected during pregnancy in nearly 1/3 of cases, supporting the recommendation that pregnancy is a specific time for viral controllers that ART use should be instituted and that the theoretical concerns about loss of viral control are warranted. We discuss this within this context of the discussion and following your comment on the fact that this is one of the most interesting findings, we have adjusted the wording of the final conclusion of the article to further emphasise this.
Concerning timing of determination of viral load, there was significant heterogeneity between cases, given the retrospective nature and timing of data collection, for older cases, written paper notes were sometimes the only source of these data, and date of note entry does not always accurately capture the date of sample collection. Where possible electronic pathology records were also consulted. If a women was diagnosed in pregnancy, the last determination of an undetectable viral load will have been made post-partum, off ART, as per the eligibility criteria for case inclusion. For maximum transparancy we have included all viral load measurements recorded during pregnancy in the summary table of cases, alongside gestation at which ART was commenced in supplementary table 2.
We believe highlighting how UK guidelines have changed over the study time in the discussion, as the main driver for differences in management practice, is an important contextual consideration. Given their rarity, in the UK, we suggest that such cases are managed by expert teams in tertiary clinical care and at Imperial College London we are the national centre. Our latest publication, Khan et al. 2023, referenced in the text, describes how enhanced immunological, serological and molecular assays can be of use in determining evidence of low level viral replication and immune activation to guide treatment decisions in non-pregnant viral controllers. Here we attempt to characterise pregnant viral controllers, where data were available, in the same manner i.e. as molecular positive or negative and with or without evidence immune activation, hence the reference to pro-viral DNA and CD4/CD8 ratios.
We think that the low level of PTB in the context of likely low systemic HIV driven immune activation is an important observation given the known immune mechanisms and role of ART exposure underlying this birth outcome.
We thank the reviewer for noting that we had not defined RCT in the first paragraph and have adjusted this.
Reviewer 3 Report
Comments and Suggestions for Authors
The reviewed manuscript aims to provide cumulative data over a 20-year experience on the management of pregnancy in HIV viral controllers. Overall, the manuscript presents a nicely designed work and is well-written, and the presented data would be of significant interest to the field. I only have a few minor comments and/or suggestions to be considered that I believe will strengthen the manuscript and improve its impact.
Specific comments and suggestions to consider:
· I believe that the abstract does not need to be stratified in sections, i.e background, material & methods etc
· In the introduction (lines 67-68), it has been proven that undetectable plasma VL equals untransmittable even in unprotected sexual intercourse. Please refer to the UK-led PARTNER I and II studies.
· The inclusion criteria included HIV viral load less than 100 copies/mL on at least two consecutive assays. Perhaps the authors mean two consecutive measurements, and not HIV VL assay. Also, how was the plasma VL data from the Amplicor HIV-1 Monitor 1.5 sensitive assay (with LLoQ = 400) used to include or exclude a subject from the study population?
· Co-morbidities and complications should include additional STDs, such as syphilis, gonorrhoea, chlamydia etc
· Results are analyzed with plasma VL below vs. above 50 cp/ml, which conflicts with the inclusion criteria.
Author Response
Thank you for your supportive review and request for minor edits to strengthen the manuscript.
With reference to the abstract headings, we have followed the journal template.
We have added the UK PARTNER I study in the references for U=U, thanks for the recommendation
With regards the inclusion criteria of a VL < 100 copies/mL and results reported >50 copies; the inclusion criteria were made to reflect differences in LLOQ of HIV RNA tests over time, see supplementary table 1, to maximise eligibility across the regional network and significant retrospective study period. Thank you for highlighting the Amplicor Monitor 1.5 assay has a LLOQ of 400 copies/mL. In cases in which this assay was used, a confirmatory assay on at least two other platforms with a LLOQ < 100 copies/mL off ART was available, hence such cases could be included.
With increasing sensitivity of assays over the years and evidence base for a clinical threshold of ‘undetectable’ as < 50 copies/mL, we decided to report the results in this way. In our recent manuscript characterising the national IDRIS cohort (Khan et al 2023), and in clinical practice with assays with even lower LLOQ, we are increasingly observing very low level viraemia of uncertain significance, which could be important to detect in viral controllers as a potential early indication of loss of control.
Thank you for suggesting complications should include STIs, we would agree, however none were reported in the data collection forms.
Round 2
Reviewer 2 Report
Comments and Suggestions for Authors
Thank you for your revision